



# Atmospheric aerosol, gases and meteorological parameters measured during the LAPSE-RATE campaign

David Brus[1], Jani Gustafsson[1], Osku Kemppinen[2,†], Gijs de Boer[3,4], and Anne Hirsikko[1]

[1]Finnish Meteorological Institute, Erik Palménin aukio 1, P.O. Box 503, FIN-00100 Helsinki, Finland
[2]Kansas State University, Department of Physics, 1228 N. 17th St., 66506, Manhattan, Kansas, USA
[3]University of Colorado, Cooperative Institute for Research in Environmental Sciences, 216 UCB, 80309, Boulder, Colorado, USA
[4]National Oceanic and Atmospheric Administration, Physical Sciences Laboratory, 325 Broadway, 80305, Boulder, Colorado, USA
[†]Currently at University of Maryland, Earth Systems Science Interdisciplinary Center, 5825 University Research Ct suite 4001, College Park, MD 20740

**Correspondence:** David Brus (david.brus@fmi.fi)

**Abstract.** Small Unmanned Aerial Systems (sUAS) are becoming very popular as affordable and reliable observation platforms. The Lower Atmospheric Process Studies at Elevation - a Remotely-piloted Aircraft Team Experiment (LAPSE-RATE), conducted in the San Luis Valley of Colorado (USA) between July 14th - 20th, 2018, gathered together numerous sUAS, remote sensing equipment and ground based instrumentation. Flight teams from the Finnish Meteorological Institute and the

Kansas State University co-operated during LAPSE-RATE to measure and investigate the properties of aerosol particles and gases at the surface and in the lower atmosphere. During LAPSE-RATE the deployed instrumentation operated reliably, resulting in a scientifically sound observational dataset. Our observations included aerosol particle number concentrations and size distributions, concentrations of $CO_2$ and water vapor, and meteorological parameters.

All data sets have been uploaded to the Zenodo LAPSE-RATE community archive (https://zenodo.org/communities/lapse-rate/).
The dataset DOIs for FMI airborne measurements and surface measurements are available here:
https://doi.org/10.5281/zenodo.3993996, Brus et al. (2020a), and for KSU airborne measurements and surface measurements are available here: https://doi.org/10.5281/zenodo.3736772, Brus et al. (2020b).

## 1    Introduction

The concentration of aerosol particles released from primary sources or formed by gas-liquid conversion and various trace gases
characterize the air quality worldwide. At the same time, planetary boundary layer (PBL) mixing height and lower atmospheric wind speed and direction are influenced by a variety of factors (e.g. solar and surface stored energy and terrain inhomogeneity, Carbone et al. (2010)). Removal of aerosol particles and gases from the atmosphere depends mostly on dispersion rates, transport, wet deposition and other atmospheric dynamical properties (Tunved et al., 2013). To understand the formation and removal of particles and gases, it is therefore necessary to measure their properties and the atmospheric conditions supporting
those properties throughout the vertical profile. In-situ observation of PBL properties are obtained through various techniques,





including balloon soundings, tethersondes, dropsondes and hot-air balloons (e.g. Laakso et al., 2007; Greenberg et al., 2009; Nygård et al., 2017), towers (e.g. Heintzenberg et al., 2011; Andreae et al., 2015), and recently, by Unmanned Aerial Systems (UASs) (e.g. Ramanathan et al., 2007; Jonassen et al., 2015; Kral et al., 2018; Barbieri et al., 2019; de Boer et al., 2020a).

Land-air interactions, including influences of continental, inhomogeneous terrain on lower atmospheric thermodynamic and kinematic states and the vertical distribution of aerosol and gas properties, were investigated during The Lower Atmospheric Process Studies at Elevation - a Remotely-piloted Aircraft Team Experiment (LAPSE-RATE) field campaign. LAPSE-RATE took place in the greater San Luis Valley (SLV), Colorado between July 14th-20th, 2018. The campaign was organized in conjunction with the 6th Conference of the International Society for Atmospheric Research using Remotely-piloted Aircraft (ISARRA, de Boer et al., 2018). Daily operational plans were developed and executed to observe several atmospheric phenomena, including boundary layer evolution during morning hours, the diurnal cycle of valley flows, convective initiation, and the properties of gases and aerosol particles (de Boer et al., 2020b). LAPSE-RATE flights were conducted under both Federal Aviation Administration (FAA) Certificates of Authorization (COAs) and FAA Part 107, with the COAs allowing maximum flight altitudes of 914 m above the ground level. In addition to the airborne assets, a variety of ground-based observational assets were deployed (see de Boer et al., 2020b; Bell et al., 2020, for details). The FMI-KSU flight team consisted of two operators from the Finnish Meteorological Institute (FMI) and one operator from the Kansas State University (KSU). The FMI-KSU team targeted the LAPSE-RATE scientific theme of aerosol properties during most flight days, except on July 19th when the team joined the common effort to focus on cold-air drainage flows that set up during the night-time.

A variety of primary and secondary aerosol particle sources in the SLV, including agriculture, the Great Sand Dunes National Park, wildfires, biogenic emissions, and large-scale advection of particles from the deserts and mountainous regions of the Western United States, made this an interesting location to characterize aerosol properties and their variability. Coupled with the substantial diurnal cycles in atmospheric temperature, humidity, turbulence and thermodynamic mixing, as well as the frequent occurrence of thunderstorms in the mountains surrounding the SLV, aerosol properties were found to vary over the course of the week, as documented in Brus et al. (2020c). Routine lower-atmospheric profiling allowed our team to document particle sizes and concentrations, including the occurrence of new particle formation events, and the connection of these phenomena to boundary layer and synoptic wind regimes.

## 2  Instrument and Platform Description

The FMI team deployed two rotorcraft (FMI-PRKL1 and FMI-PRKL2) during LAPSE-RATE. Both rotorcraft are custom-built around the Tarot X6 hexacopter frame, which is 960 mm in diameter (rotor-to-rotor). The maximum endurance of these rotorcraft was about 15 min using 22.2V, 16000 mAh rechargeable lithium polymer (LiPo) batteries. The maximum take-off weight for these rotorcraft was 11 kg. Flights were carried out using a 3DR Pixhawk PX4 flight controller with the Ardupilot software, in a manual (stabilized) mode for FMI-PRK1 and loiter (GPS position fix) mode for FMI-PRKL2. The same propulsion system, consisting of 340 kV brushless motors, 40A Electronic Speed Controllers and 18 inch (5.5 inch pitch) carbon fiber



propellers, was used for both rotorcraft. The rotorcraft's setup allowed for lifting approximately 2 kg of active payload (i.e. scientific instrumentation).

The first rotorcraft (FMI-PRKL1) was equipped with a particle measurement module consisting of two condensational particle counters (CPC, model 3007, TSI Corp.), a factory-calibrated optical particle counter (OPC, model N2, Alphasense) and a meteorological Arduino breakout (Bosh BME280, P, T and RH). Each CPC was calibrated at FMI to a different cut-off diameter, 7 and 14 nm, respectively. Such a configuration allows for the observation of freshly nucleated particles in a diameter range of 7 to 14 nm (see e.g. Altstädter et al., 2015, 2018) . The voltage applied to a thermal electric device (TED) of the

CPC corresponds directly to the temperature difference between the saturator and condenser and determines how fast particles grow to CPC detectable sizes. The TED values of 2000 and 1000 mV were used for CPC1 and CPC2, respectively, during the campaign. The particle module was covered from all sides except the bottom with a polylactide (PLA) foam cover to protect the sensors from solar radiation and keep the particle module thermally stable. The BME280 sensor was located below the particle module and was shielded from solar radiation, but not forcefully aspirated.

The second rotorcraft (FMI-PRKL2) was equipped with a gas module, consisting of a flow-through $CO_2$ concentration sensor (Carbocap model GMP343, Vaisala Inc.), a $CO_2$ and water vapor analyzer (model Li-840A, Li-Cor Environmental) and a sensor for measuring concentrations of CO, $NO_2$, $SO_2$, and $O_3$ (model AQT400, Vaisala Inc.). A meteorological Arduino breakout (Bosh BME280) was used to measure P, T and RH. The BME280 sensor was mounted identically to the BME280 on the FMI-PRKL1.

Both $CO_2$ sensors were forcefully-aspirated using micro-blowers configured as air pumps (Murata, model MZB1001T02) connected to the exhaust of the sensors with flow rate of 0.6 L.min$^{-1}$. A Gelman filter was placed in the sample airstream in front of both $CO_2$ sensors to avoid contamination of the optical path.

Unfortunately, we were not able to acquire vertical profiles neither ground level concentrations of gases with the Vaisala AQT400 sensor during LAPSE-RATE. All recorded concentrations of gases ($NO_2$, $O_3$, CO and $SO_2$) were far below the

manufacturer declared detection limits, that is why no data set for the Vaisala AQT400 sensor is provided.

A third FMI module was operated on the ground. It consisted of a condensational particle counter (CPC, model 3007, TSI Corp.), an optical particle counter (OPC, model N2, Alphasense) and a TriSonica Mini Weather Station (Applied Technologies, Inc.). The surface sensor module was covered from all sides with PLA foam to protect sensors from solar radiation and keep the particle module thermally stable. It was placed on the roof top of a car at about 2 m from the ground, the TriSonica Mini

was mounted on a 45 cm long carbon fiber tube on top of the surface sensor module, i.e. about 2.75 m from the ground. Data of both FMI rotorcraft and the FMI surface sensor module were logged separately to embedded Raspberry Pi 3+ minicomputers using Python scripts to produce ASCII comma separated files, which were later converted to NetCDF format.

The KSU used a DJI Matrice 600 Pro rotorcraft without any modifications beyond the payload attachments, with the aircraft controlled with the Matrice 600 Pro remote controller. Both DJI TB47S (6x4500 mAh, 22.2 V) and DJI TB48S (6x5700 mAh,

22.8 V) rechargeable lithium polymer batteries were used, alternating between flights. The maximum gross take-off weight recommended by the manufacturer is 15.5 kg resulting in a maximum payload capacity of roughly 5.5 kg. The Matrice 600 Pro of KSU was equipped with an optical particle counter (POPS, Handix Scientific LLC). During LAPSE-RATE the POPS was



used both with a horizontally oriented naked inlet with inner diameter of 1.7 mm (0.069 inches), and a vertically oriented tube inlet of approximately 45 cm (18 inches) long and with an inner diameter of 3.175 mm (0.125 inches). POPS included custom

electronics, and logged data to an onboard microSD card. POPS was attached to the top surface of the rotorcraft body with Velcro tape. It was not shielded from direct sunlight during the flights, but was kept in shade while on the ground. A duplicate POPS instrument was operated as a ground reference and was located approximately 1.8 meters above the ground level. The overview of FMI-KSU team instrumentation and their operational characteristics are summarized in Table 1.

## 3   Description of measurement location, flight strategies and completed sampling

As mentioned above, the San Luis Valley provided a variety of aerosol particle sources, sinks and processing modes. In combination, the variability resulting from this processing made the San Luis Valley an interesting place to study aerosol properties and their spatial and temporal variability. One of the main sources of income for San Luis Valley inhabitants comes from farming and ranching. The primary crops grown here include potatoes, alfalfa, native hay, barley, wheat, quinoa and vegetables like lettuce, spinach, and carrots. Crops are irrigated by surface flooding water but mainly by center pivot sprinklers.

Uncultivated land is often covered with low brush such as rabbitbrush, greasewood and other woody species. The land is also heavily used for grazing. The soils of the SLV are generally coarse, gravelly, sandy soils or loam, and derived mainly from volcanic rocks (Lapham, 1912).

The FMI-KSU team operated from one location throughout the entire LAPSE-RATE campaign. This location was situated along County Road 53, approximately 15 km north from Leach Airport (37°54'32.94" N, 106°2'6.83" W, 2291 m MSL), see

Fig. 1. The location was flat and treeless, surrounded by grazing farmlands and generally very quiet. Occasionally the site experienced emissions and aerosol production from the operation of local farm trucks. The FMI team was permitted to operate to a maximum altitude of 914 m AGL under a FAA COA, however the aircraft was only flown to a maximum altitude of 893 m AGL. All flights operated by the KSU team were conducted up to a maximum altitude of 121 m AGL, the maximum altitude permitted under FAA Part 107.

During July 15th-18th the FMI –KSU team conducted missions focusing on profiling of aerosol particle and gas properties. In total, the FMI team completed 38 vertical profile flights: 14 flights with the particle module and 24 flights with the gas module (see Table 2). The KSU team completed a total of 33 flights with their payload, including 40 individual vertical profiles. It should be noted that some of these KSU profiles were redundant, made within a few minutes of each other, and repeated in the exact same location as another profile. These redundant flights were completed to test the instrument consistency. The FMI

flight strategy was to conduct only vertical flights and reach as high an altitude as possible in very short time. FMI ascent rates were between 5-8 and 3-5 m.s$^{-1}$ and descent rates were about 2-5 and 2-3 m s$^{-1}$ for flights with the particle module (FMI-PRKL1) and gas module (FMI-PRKL2), respectively. Vertical profiling was performed in cycles by alternating flights with the aircraft carrying the FMI particle module, the FMI gas module and the KSU platform, with about 30 minutes in between flights (please see Tables 2 - 4 for details on timing, frequency and achieved altitude of all flights). The FMI surface module and KSU

surface POPS were logging continuously during the time periods of flight operations (see Table 5 and 6 for details).



On the last day of FMI-KSU operations during LAPSE-RATE (July 19[th]), the team joined the common effort to evaluate cold-air drainage from local valleys during the morning hours. The operation started about 11:45 UTC (5:45 a.m. local time) with vertical profiles every 30 min lasting about 5 hours. Only the FMI-PRKL2 and KSU rotorcraft took part in these cold-air drainage flights in an alternating fashion. For these flights ascent and descent rates were reduced to about 2 m s$^{-1}$ given that the
target maximum altitude during these flights only extended to 350 m AGL.

## 4 Data Processing and Quality Control

Data files generated by the FMI particle module (FMI-PRKL1) were formatted in NetCDF format and were named according to the general naming convention for the LAPSE-RATE files (FMI.PRKL1.a1.yyyy.mm.dd.hh.mm.ss.cdf), as outlined in de Boer et al. (2020b). Missing data in the dataset are marked as -9999.9. These files provide information at 1 Hz, and include
the Raspberry Pi RTC time stamp, aircraft location (GPS latitude and longitude in degrees and altitude in MSL meters), basic meteorology, including temperature (°C), pressure (hPa) and relative humidity (%). Additionally included in these data files are the the FMI-PRKL1 aircraft attitude data (pitch, roll, yaw and heading in degrees, GPS ground speed in m s$^{-1}$ and vertical ascent rate, also in m s$^{-1}$). These variables are published as they were measured without any corrections. Finally, these files include the total particle concentration measured by the pair of CPCs (model 3007, TSI Inc.). As a reminder, these two CPCs
had different cut-off diameters ($D_{50}$) at 7 and 14 nm (the lower size boundaries), with the upper boundary particle size more than 1 μm . These quantities are also provided in the files as they were measured, with no corrections applied. NetCDF files created from the measurements of the OPC-N2 particle counter were saved separately under the file name FMI.PRKL1OPC-N2.a1.yyyy.mm.dd.hh.mm.ss.cdf. These files contain 0.5 Hz resolution data, including the time stamp, aircraft location (GPS latitude and longitude in degrees and altitude in meters, total aerosol number concentration in cm$^{-3}$ and total aerosol volumetric
concentration in μm$^3$.cm$^{-3}$, both in a size range of 0.38-17 μm). Also included are particle number concentrations (cm$^{-3}$) in each bin, including measurements in 16 size bins with mid-bin diameters of 0.46, 0.66, 0.92, 1.19, 1.47, 1.83, 2.54, 3.5, 4.5, 5.75, 7.25, 9, 11, 13, 15 and 16.75 μm, the calculated d$N$/dlog$D_{\mathrm{p}}$ (cm$^{-3}$) values in each size bin, and measured PM1, PM2.5 and PM10 mass concentrations in μg.m$^{-3}$, respectively.

The FMI gas module dataset (FMI.PRKL2.a1.yyyy.mm.dd.hh.mm.ss.cdf) includes NetCDF files containing 1 Hz data. Miss-
ing data in the dataset are marked as -9999.9. These files include the Raspberry Pi RTC time stamp, aircraft location (GPS latitude and longitude in degrees, and aircraft altitude in MSL meters), and basic meteorology including temperature in degrees °C, pressure in hPa and relative humidity in %. All of these data are published as they were measured without any corrections or additional quality control. In addition, these files contain the PRKL2 aircraft attitude data (pitch, roll, yaw and heading in degrees, GPS ground speed in m s$^{-1}$, and vertical ascent rate in m s$^{-1}$). Also included are the measured $CO_2$ concentration
in parts per million (ppm) and dew point temperature in °C as measured by Licor Li-840A. The $CO_2$ concentrations (ppm) measured by the Vaisala GMP343 probe are reported at 0.5 Hz. The $CO_2$ concentrations measured by the Licor Li-840A are internally compensated for water vapor concentration, pressure changes and atmospheric temperature (for details please see the Licor Li-840A Instruction manual). Data from the Vaisala GMP343 sensor were compensated for pressure, temperature, RH





(obtained from the BME280 sensor) and oxygen in post-processing by using the proprietary compensation algorithm provided

by Vaisala. Further, laboratory-derived calibration constants were applied to the data sets collected by both sensors. Both $CO_2$

sensors were calibrated in the laboratory before and after LAPSE-RATE and showed no drift in calibration. The sensors were

calibrated against standard carbon dioxide gases (traceable to WMO $CO_2$ scale X2007 at the FMI) at several concentrations

(zero gas and 436 ppm before the campaign, and 370, 405.4 and 440.2 ppm after the campaign). Also, both sensors were

tested in the lab against a calibrated, high-precision gas concentration analyzer (Picarro model G2401, Picarro, Inc.) at ambient

$CO_2$ concentration. The GMP343 data were biased on average -3.4 (±1.3) ppm and the Licor 0.3 (±1.1) ppm. The dew point

measurement from the Licor LI-840A was calibrated against a DewMaster Chilled Mirror Hygrometer (Edgetech Instruments

Inc.).

Measurements collected by the FMI ground module (NetCDF file names FMI.GROUNDTRISONICA-CPC.a1.

yyyymmdd.hhmmss.cdf) are saved in files that include data at 1-minute resolution, including the Raspberry Pi RTC time stamp,

wind speed ($m.s^{-1}$), wind direction (deg), temperature (°C), relative humidity (%), and pressure (hPa) measured by the Trisonica

mini weather station. Also saved is the total aerosol number concentration ($cm^{-3}$) measured by CPC (model 3007, TSI Inc.).

The second ground module dataset (FMI.GROUNDOPC-N2.a1.yyyymmdd.hhmmss.cdf) also includes 1-minute resolution

data of the Raspberry Pi RTC time stamp, total aerosol number concentration ($cm^{-3}$), total aerosol volumetric concentration

($\mu m^3.cm^{-3}$) in a size range of 0.38-17 µm. Also included are the particle number concentrations in each bin (16 size bins),

calculated d$N$/dlog$D_p$ ($cm^{-3}$) values in each bin (16 size bins), and the measured PM1, PM2.5 and PM10 mass concentrations

($\mu g.m^{-3}$) respectively. Missing data in both of these datasets are marked as -9999.9.

The KSU airborne dataset (KSU.M600POPS.a1.yyyymmdd.hhmmss.cdf) includes 1 Hz measurements including the POPS

internal RTC time stamp, aircraft location (GPS altitude (m MSL), altitude (m AGL), latitude and longitude (deg)), total aerosol

particle count and total aerosol particle concentration ($cm^{-3}$) in the size range 0.13–3 µm. Also included are atmospheric

pressure (hPa), internal air flow ($cm^3.s^{-1}$), counts per bin (16 size bins) and calculated d$N$/dlog$D_p$ ($cm^{-3}$) values in each bin (16

size bins).

Additionally, the KSU ground dataset (KSU.SURFACEPOPS.a1.yyyymmdd.hhmmss.cdf) includes 1 Hz resolution data of

POPS internal RTC time stamp, total aerosol particle count in the size range 0.13–3 µm, total aerosol particle concentration

($cm^{-3}$) in the size range 0.13–3 µm, pressure (hPa), internal air flow ($cm^3.s^{-1}$), counts per bin (16 size bins) and calculated

d$N$/dlog$D_p$ ($cm^{-3}$) values in each bin (16 size bins with mid-bin diameters: 1.38, 1.51, 1.66, 1.83, 2.00, 2.19, 2.50, 2.97,

3.99, 5.34, 8.02, 11.52, 14.66, 18.93, 25.06 and 32.56 µm). All variables are published as they were measured without any

corrections.

Throughout the collected data set, there were occasional peaks in detected aerosol number concentration caused by farm

vehicles passing our sampling location. Some of these peakes resulted in particle counts up to 40 000 $cm^{-3}$ from the CPC and

up to 16 000 $cm^{-3}$ from the POPS. These peaks typically lasted approximately 3 minutes, and were removed from the datasets.

Since hysteresis in T and RH profiles collected by both FMI aircrafts was noticeable, we recommend to use only data from the

ascending portion of the flown profiles. Data from the Vaisala GMP343 probe suffer from an inaccurate pressure compensation

algorithm provided by Vaisala. Therefore, we recommend use of Licor-840A data over those from the Vaisala GMP343 probe.





The merged data sets on meteorology, aerosol and gases concentrations are presented in Fig. 2 and Fig. 3 for airborne and surface measurements respectively.

## 5 Data Availability

Data sets collected by Finnish Meteorological Institute and Kansas State University during LAPSE-RATE were published together under the LAPSE-RATE community at Zenodo open data repository. For data collected using the platforms outlined in this paper, the Finnish Meteorological Institute FMI-PRKL1, FMI-PRKL2 and FMI surface module data sets can be all found here: https://doi.org/10.5281/zenodo.3993996, Brus et al. (2020a). The Kansas State University KSU M600 and KSU surface data sets can be found both here: https://doi.org/10.5281/zenodo.3736772, Brus et al. (2020b).

All data sets have undergone preliminary quality control and false readings were eliminated. All files are available in netCDF format for each individual UAS flight and the surface data sets are available as separate daily files.

## 6 Summary

This publication summarizes measurements collected and data sets generated by the FMI and KSU teams during LAPSE-RATE. LAPSE-RATE took place in the San Luis Valley of Colorado during the summer of 2018. In section 2, we provided an overview of the rotorcraft deployed by these teams during this campaign, and offer insight into the types of payloads that were deployed. In Section 3 we described the teams' scientific goals and flight strategies while section 4 provides a glimpse at the data sets obtained, including a description of the measurement validation techniques applied. Section 5 provided information on data sets availability. To our knowledge, the data collected by these flights represent some of the only known profiles of aerosol properties collected over the San Luis Valley.

While the overall data set is limited in temporal and spatial coverage, these measurements offer unique insight into the vertical structure and diurnal variability of atmospheric thermodynamic, aerosol and gas parameters in a new location. Such observations would be difficult to obtain with other profiling methods (such as ground-based remote-sensing instrumentation). The temporal frequency and spatial extent covered by these profiles is unique to data collected using unmanned aircraft systems. This data set offers a nice example of how these systems can help to inform our general understanding of these parameters, which can be very important to understanding weather anc climate.

Finally, work to analyze these data is ongoing. A description of the vertical structure of key parameters as observed during LAPSE-RATE has been submitted for publication (see Brus et al., 2020c), and includes evidence of new particle formation events occurring over the San Luis Valley. Additional work is planned to assess the thermodynamic and aerosol particle properties associated with the cold-air drainage flows that were found to occur during the night in the valley.

*Author contributions.* D.B. and G.B. planned and coordinated FMI missions during LAPSE-RATE campaign, D.B. conducted particle module measurements, processed, analyzed and QC FMI dataset, written the manuscript. J.G. provided technical support during the campaign,



conducted measurements with gas module. A.H. wrote the manuscript and QC FMI dataset. O.K. conducted all KSU flights. G.B. and D.B.
220  prepared and QC KSU dataset. All authors edited the manuscript.

*Competing interests.*  The authors declare no conflict of interest.

*Acknowledgements.*  The authors would like to acknowledge the following financial support of this effort: KONE foundation, ACTRIS-2 -
the European Union's Horizon 2020 research and innovation programme under grant agreement (No 654109), ACTRIS PPP - the European
Commission under the Horizon 2020 – Research and Innovation Framework Programme, H2020-INFRADEV-2016-2017 (Grant Agreement
225  number: 739530), Academy of Finland Center of Excellence programme (grant no. 307331) and the US National Science Foundation CA-
REER program (1665456). In addition, limited general support for LAPSE-RATE was provided by the US National Science Foundation
(AGS 1807199) and the US Department of Energy (DE-SC0018985) in the form of travel support for early career participants. Support for
the planning and execution of the campaign was provided by the NOAA Physical Sciences Laboratory and NOAA UAS Program Office.
Finally, the support of UAS Colorado and local government agencies (Alamosa County, Saguache County) was critical in securing site per-
230  missions and other local logistics. D.B. and J.G. would like to especially acknowledge Dave L. Coach for acting as a Pilot-In-Command for
the FMI team. Handix Scientific, LLC is acknowledged for providing their POPS instruments for the campaign at no cost.





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





**Figure 1.** A) Map of wider San Luis Valley, CO, B) map cutout of FMI-KSU team location - 15 km north of Leach airport, located approximately 3.2 km ENE of the commercial district of Center, Colorado and 32 km NNW of Alamosa, Colorado. ©Google Maps C) FMI-KSU team operation spot NE view, D) FMI-KSU team operation spot view towards E, E) view of typical surroundings of FMI-KSU team spot.

Wehner, B., Siebert, H., Ansmann, A., Ditas, F., Seifert, P., Stratmann, F., Wiedensohler, A., Apituley, A., Shaw, R. A., Manninen, H. E. and

Kulmala, M.: Observations of turbulence-induced new particle formation in the residual layer, Atmos. Chem. Phys., 10(9), 4319–4330, doi:10.5194/acp-10-4319-2010, 2010.

Wehner, B., Werner, F., Ditas, F., Shaw, R. A., Kulmala, M. and Siebert, H.: Observations of new particle formation in enhanced UV irradiance zones near cumulus clouds, Atmospheric Chemistry and Physics, 15(20), 11701–11711, doi:10.5194/acp-15-11701-2015, 2015.



**Figure 2.** Overview of data collected in vertical profile, mean values with standard deviations as error-bars: A) temperature, B) relative humidity, C) pressure, D) total aerosol number concentration by CPCs and POPS (0.13–3 μm) respectively, E) aerosol number concentration by OPC-N2 (0.3-18μm) and F) $CO_2$ concentration by Licor Li-840A and Vaisala GMP343 and dew point temperature by Licor Li-840A.



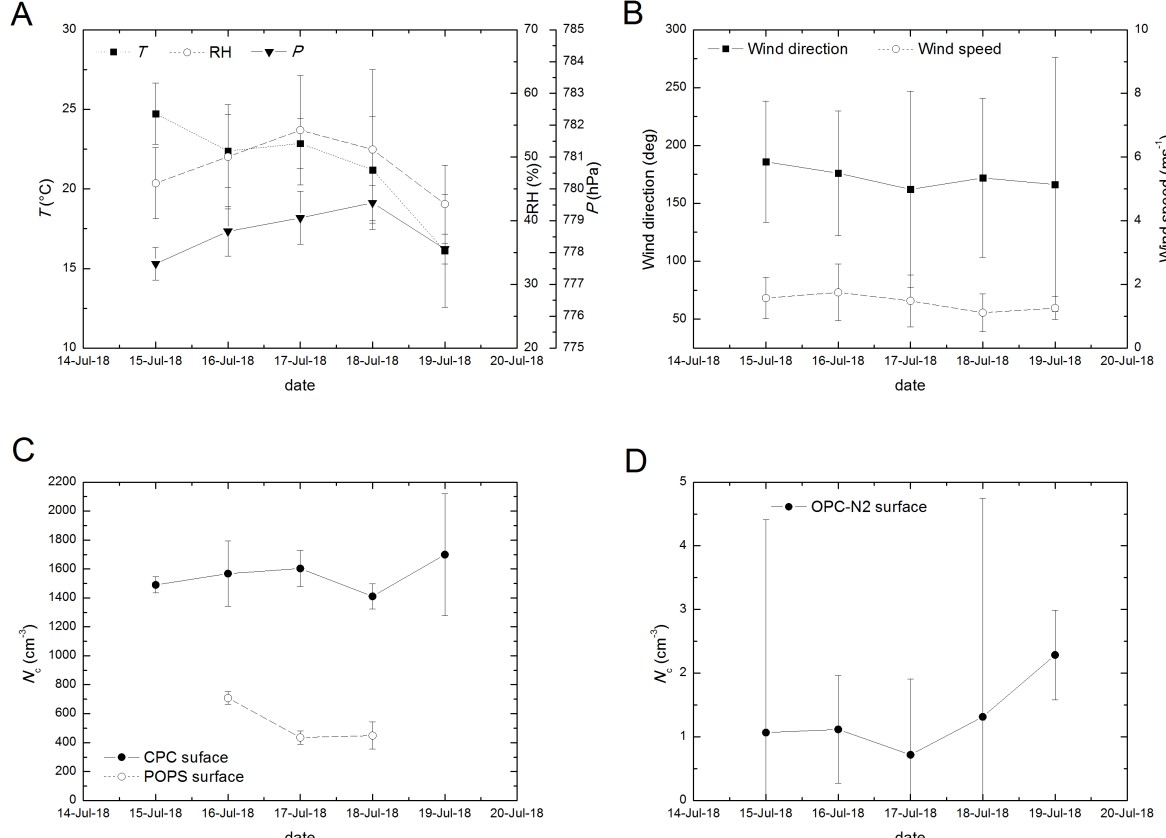

**Figure 3.** Overview of data collected at the surface, mean values with standard deviations as error-bars: A) temperature, relative humidity, pressure, B) wind direction and speed by Trisonica mini weather station C) total aerosol number concentration by CPC and POPS (0.13–3μm), and D) aerosol number concentration by OPC-N2 (0.3-18μm)

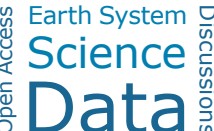

**Table 1.** An overview of sensors and their operational characteristics use during airborne and ground based measurements.

| Sensor | Resolution | Accuracy | Range | Flow rate | Response time |
|---|---|---|---|---|---|
| **CPC, TSI 3007** | | | | | |
| Particle conc. (cm$^{-3}$) | | | 0-10$^5$ cm$^{-3}$, 0.01->1 μm | 0.7 L.min$^{-1}$ | 1 s |
| **OPC, Alphasense N2** | | | | | |
| Particle conc. (cm$^{-3}$) | | | 0-10$^4$ part.s$^{-1}$, 0.38-17 μm at 16 bins | 1.2 L.min$^{-1}$ | 1 s |
| **BME280** | | | | | |
| T (°C) | 0.01 | ±0.5 °C | -40-85 °C | | 1 s |
| RH (%) | <0.01 | ±3 % | 0-100 % | | 1 s |
| Pressure (hPa) | 0.18 Pa | ±1 hPa | 300-1100 hPa | | 6 ms |
| **GMP343, Vaisala** | | | | | |
| CO$_2$ (ppm) | 14-bits | ±3ppm +1% of reading | 0-1000 ppm | 0.6 L.m$^{-1}$ | 2s |
| **Li-840A, Licor** | | | | | |
| CO$_2$ (ppm) | 14-bits across user-specified range | <1.5% of reading | 0-20000 ppm | 0.6 L.m$^{-1}$ | 1s |
| T$_d$ (°C) | 14-bits across user-specified range | <1.5% of reading | -25-45 °C, RH: 0 to 95% RH Non-Condensing | 0.6 L.m$^{-1}$ | 1s |
| **TriSonica Mini WS, Applied Tech.** | | | | | |
| T (°C) | 0.1 | ±2°C | -25-80 °C | | 1 s |
| RH (%) | 0.1 | ±3% | 0-100 % | | 1 s |
| Pressure (hPa) | 1 | ±10 hPa | 500-1150 hPa | | 1 s |
| Wind speed (m.s$^{-1}$) | 0.1 | ±0.1 m.s$^{-1}$ | 0-30 m.s$^{-1}$ | | 1 s |
| Wind direction (deg.) | 1 | ±1 deg. | x/y: 0-360 deg., z: ±30 deg. | | 1 s |
| **POPS, Handix Sci.** | | | | | |
| Particle conc. (cm$^{-3}$) | | ±10% <1000 cm$^{-3}$ at 0.1 L.m$^{-1}$ | 0-1250 cm$^{-3}$, 0.13 – 3 μm at 16 bins | 0.18 L.min$^{-1}$ | 1 s |

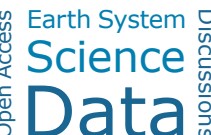

**Table 2.** FMI-PRKL1 - particle module, an overview of vertical profile flight times and achieved altitude

| platform | date | start time (UTC) | end time (UTC) | altitude (MSL) | altitude (AGL) |
|---|---|---|---|---|---|
| **FMI-PRKL1** | 15/07/18 | 16:51:49 | 16:56:55 | 2483.49 | 175.49 |
| | | 17:24:40 | 17:32:00 | 2436.60 | 128.60 |
| | | 18:24:54 | 18:33:12 | 2449.68 | 141.68 |
| | 16/07/18 | 15:06:47 | 15:14:55 | 2819.67 | 511.67 |
| | | 16:20:53 | 16:29:21 | 2823.44 | 515.44 |
| | | 17:30:33 | 17:41:14 | 2730.81 | 422.81 |
| | | 18:37:31 | 17:47:36 | 2949.73 | 641.73 |
| | | 20:33:20 | 20:42:56 | 2994.55 | 686.55 |
| | 17/07/18 | 18:17:18 | 18:26:03 | 2924.22 | 616.22 |
| | | 19:24:35 | 19:35:13 | 2992.33 | 684.33 |
| | | 20:41:37 | 20:53:53 | 3119.02 | 811.02 |
| | 18/07/18 | 14:03:55 | 14:14:33 | 3155.00 | 847.00 |
| | | 15:26:03 | 15:36:27 | 3028.59 | 720.59 |
| | | 16:59:06 | 17:05:55 | 3201.53 | 893.53 |



**Table 3.** FMI-PRKL2 - gas module, an overview of vertical profile flight times and achieved altitude

| platform | date | start time (UTC) | end time (UTC) | altitude (MSL) | altitude (AGL) |
|---|---|---|---|---|---|
| **FMI-PRKL2** | 15/07/18 | 16:02:28 | 16:06:28 | 2356.88 | 48.88 |
| | | 17:12:39 | 17:17:13 | 2441.78 | 133.78 |
| | | 17:59:24 | 18:05:11 | 2409.02 | 101.02 |
| | 16/07/18 | 17:00:44 | 17:08:28 | 2675.27 | 367.27 |
| | | 18:00:08 | 18:08:55 | 2703.60 | 395.60 |
| | | 19:57:24 | 20:06:33 | 2781.37 | 473.37 |
| | 17/07/18 | 17:37:19 | 17:46:52 | 2773.00 | 465.00 |
| | | 18:53:43 | 19:02:34 | 2814.49 | 506.49 |
| | | 20:27:20 | 20:36:53 | 2910.11 | 602.11 |
| | 18/07/18 | 13:19:00 | 13:31:21 | 2971.37 | 663.37 |
| | | 14:29:00 | 14:44:16 | 3004.27 | 696.27 |
| | | 16:25:00 | 16:40:26 | 3013.40 | 705.40 |
| | | 18:10:00 | 18:24:36 | 2912.85 | 604.85 |
| | 19/07/18 | 11:44:33 | 11:53:42 | 2669.10 | 361.10 |
| | | 12:07:00 | 12:24:09 | 2658.34 | 350.34 |
| | | 12:46:00 | 12:53:43 | 2667.16 | 359.16 |
| | | 13:08:00 | 13:19:10 | 2666.06 | 358.06 |
| | | 13:41:00 | 13:49:25 | 2667.75 | 359.75 |
| | | 14:11:00 | 14:22:36 | 2669.65 | 361.65 |
| | | 14:42:00 | 14:55:43 | 2664.22 | 356.22 |
| | | 15:30:00 | 15:45:19 | 2657.98 | 349.98 |
| | | 16:12:00 | 16:22:25 | 2658.15 | 350.15 |
| | | 16:42:00 | 16:53:32 | 2663.10 | 355.10 |



**Table 4.** KSU POPS vertical profile flight times and achieved altitude

| platform | date | start time (UTC) | end time (UTC) | altitude (MSL) | altitude (AGL) |
|---|---|---|---|---|---|
| **KSU POPS** | 16/07/18 | 14:40:46 | 14:49:51 | 2401.3 | 125.4 |
| | | 15:23:44 | 15:37:17 | 2419.4 | 142.4 |
| | | 16:32:25 | 16:41:06 | 2410.6 | 126.6 |
| | | 18:17:44 | 18:24:58 | 2416.7 | 126.4 |
| | | 19:01:43 | 19:09:54 | 2333.0 | 39.4 |
| | | 19:43:41 | 19:47:18 | 2354.6 | 52.1 |
| | | 20:17:20 | 20:22:46 | 2429.7 | 126.8 |
| | 17/07/18 | 15:10:38 | 15:24:49 | 2401.3 | 125.7 |
| | | 17:02:14 | 17:09:42 | 2401.6 | 129.0 |
| | | 17:58:12 | 18:04:40 | 2403.5 | 127.5 |
| | | 18:29:15 | 18:40:16 | 2379.9 | 100.0 |
| | | 19:15:14 | 19:26:11 | 2417.7 | 131.3 |
| | | 20:03:59 | 20:09:49 | 2424.7 | 127.4 |
| | 18/07/18 | 12:59:49 | 13:04:27 | 2397.5 | 127.4 |
| | | 13:31:21 | 13:42:29 | 2398.4 | 127.2 |
| | | 14:16:06 | 14:22:23 | 2400.8 | 131.3 |
| | | 14:58:05 | 15:02:03 | 2394.9 | 126.2 |
| | | 15:59:52 | 16:08:59 | 2402.2 | 127.2 |
| | | 16:44:06 | 16:46:37 | 2408.2 | 128.2 |
| | | 17:30:21 | 17:38:37 | 2408.4 | 126.4 |
| | | 17:59:11 | 18:03:40 | 2415.5 | 128.6 |
| | | 18:29:37 | 18:33:06 | 2417.9 | 126.5 |
| | | 18:45:25 | 18:49:16 | 2419.4 | 126.1 |
| | 19/07/18 | 12:01:01 | 12:05:05 | 2433.6 | 138.8 |
| | | 12:28:54 | 12:31:35 | 2435.9 | 142.8 |
| | | 12:59:42 | 13:02:06 | 2429.7 | 139.1 |
| | | 13:29:40 | 13:32:18 | 2427.4 | 139.0 |
| | | 13:59:47 | 14:02:37 | 2425.1 | 137.4 |





**Table 5.** FMI surface module, an overview of data coverage

| platform | date | start time (UTC) | end time (UTC) |
|---|---|---|---|
| **FMI surface module** | 15/07/18 | 15:34:35 | 19:36:35 |
| | 16/07/18 | 14:45:29 | 22:10:47 |
| | 17/07/18 | 17:01:53 | 21:08:15 |
| | 18/07/18 | 13:03:29 | 19:05:08 |
| | 19/07/18 | 11:37:20 | 16:56:36 |

**Table 6.** KSU POPS surface - an overview of data coverage

| platform | date | start time (UTC) | end time (UTC) |
|---|---|---|---|
| **KSU POPS surface** | 16/07/18 | 15:52:34 | 16:16:44 |
| | 17/07/18 | 18:17:46 | 20:31:13 |
| | 18/07/18 | 13:30:39 | 18:13:44 |