# Peer review of "Atmospheric aerosol, gases and meteorological parameters measured during the LAPSE-RATE campaign by Finnish Meteorological Institute and Kansas State University"

_Earth System Science Data, 2020_

## Referee Comment (RC1) · Anonymous Referee #1 · 3 Nov 2020

The manuscript by Brus et al. summarizes atmospheric data collected with UAS and ground-based sensors during the LAPSE-RATE campaign in Colorado in 2018. This is a rich dataset that will likely be mined for many years to come. The introduction would benefit from the inclusion of specific objectives. The summary would benefit from a discussion of what insights these data might provide, what questions might be addressed and answered, and where might future work be headed. Some ideas are presented below for improving and condensing the tables such that the data repository may be better linked to the manuscript.

L7, this sentence is a bit awkward. Let the data speak for themselves. No need to state

in the abstract that things were reliable and scientifically sound. . .

L34, the acronyms should be listed after they are defined.

L40-45, this paragraph would benefit from the inclusion of (a) specific objectives and (b) specific hypotheses to be tested.

L47, are should be were, is should be was. careful of tense throughout the paper. past tense describing work that has been done.

L55-60, consider adding information on particle size bins and sampling rates for the particle counters.

L73, neither should be for.

Section 2 should provide information on the placement of the sensors and how this placement was designed to minimize impacts of prop wash. This reviewer is particularly concerned about this salient point, given that it does not appear that the sensors were mounted above the rotary wing airframes (e.g., Nolan et al., 2018; https://www.mdpi.com/1424-8220/18/12/4448).

L95-101, provide some citations for this information please.

L140, I see sensor sampling rates in section 4, but I think these details would be better suited to a prior section. Or maybe even consider a separate section under a heading of 'sensors' where these could be separated from the description of the UAS platforms?

L184, . . .likely caused by farm vehicles. . .

L187, suffered

Figure 1, B. The blue flag waypoints are very hard to see in the image.

Figure 3 legend, make sure to add information that this was from the car mount, approximately 2m AGL, at least as I understand it.

Table 1 is beautiful! This supports the idea of a separate section on just 'sensors',

highlighting this table.

Tables 2,3, and 4 could be combined into a single UAS measurements table, with a platform designation column and a mission column. It would also be nice to have a flight/mission number listed here that would correspond to a labeled dataset. This will make it much easier for other folks to actually use these data.

Tables 5 and 6 could be combined into a single surface measurements table, with just a platform column. It would also be nice to have a sampling period/mission number listed here that would correspond to a labeled dataset. This will make it much easier for other folks to actually use these data.

---

## Referee Comment (RC2) · Anonymous Referee #2 · 14 Dec 2020

Review of Atmospheric aerosol, gases and meteorological parameters measured during the LAPSE-RATE campaign, D. Brus et al.

This manuscript presents an overview of the data acquired from copter UAVs that conducted vertical profiles of aerosols, gases and meteorological parameters during the LAPSE-RATE campaign in the summer of 2018. Members from the Finnish Meteorological Institute (FMI) and Kansas State University (KSU) conducted measurements over a period of five days in the San Luis Valley, CO. The FMI copter measurements include vertical profiles up to nearly 900 m AGL of aerosol number concentrations and size distributions, CO2, and meteorological parameters (pressure, temperature, relative humidity), while KSU copter measurements conducted vertical profiles (up to 150 m AGL) of aerosol size and number distributions. In addition, FMI and KSU conducted ground-based measurements of aerosol and meteorological parameters to compare with the airborne copter measurements.

The data from these flights is readily available from the websites provided in the manuscript.

It is understood that ESSD is dedicated to the publication of datasets; however, key features of the dataset as well as its limits are needed to highlight the utility of the data in future publications. Several issues are described below that the authors should address before publication.

General comments: The authors allude to a number of regional / local sources and meteorological patterns that impact diurnal cycles, changes in aerosol properties, generation of new particle formation events. However, few specific examples were highlighted in the text (farm vehicles), and summary (new particle formation). A few sentences on the defining characteristics of this dataset (for example, temporally with respect to meteorological patterns and vertically with respect to atmospheric structure) would be useful.

Airborne aerosol measurements are a challenge – especially for measurements of particles larger than several micrometers in diameter in a non-isokinetic flow. A description of the KSU inlet has been provided; however, the orientation with respect to the wind and propeller wash is not clear. The stated largest diameter for the N2 and POPS are 17 and 33 um, respectively. The authors need to provide an assessment of sampling biases related to super-micron aerosol particles. I also suggest that authors compare ground-based and airborne measurements for the OPC-N2 and POPS at a range of sizes between 0.3 and 30 um diameter.

Along the same line, when comparing the OPC-N2 between the copter and the ground-based measurements (Figures 2D,E and Figures 3C,D), it appears that the N2 concen-

trations on the copter are systematically at least a factor of two larger than the ground-based measurements. Was there any additional flow control or flow measurements for the OPC-N2? The off-the-shelf version does not provide precise measurement of the air volume for determining the number concentration.

The authors state that hysteresis between ascent and descent profiles was significant. This difference is expected given the high ascent rates (up to 8 m. s-1) and considerably slower descent rates (as low as 2 m.s-1). The authors then suggest using the ascents for the best representation of the vertical profiles with no justification. Given that the ascent rates were faster, the impact of hysteresis on the vertical profile should also be greater. Are there other factors that need to be considered? What is the bias related to the hysteresis? I also suggest showing a figure to illustrate the impact of hysteresis on the vertical profiles.

The figures show time series of averaged values and variability for each flight. However, there are no examples of vertical profiles and no specific comparisons / validations between airborne and ground-based data. Yet, there are some differences between the airborne and ground-based averages that cannot be reconciled in the figures that have been shown. For example, ground-based temperature and relative humidity show no consistent relationship with the airborne observations. I would have expected to see the ground-based temperature similar to the warmest airborne temperature in a well-mixed boundary layer. As mentioned above, the ground-based number concentrations reported by the OPC N2 are consistently less than the airborne values by more than a factor of two. Otherwise, ground-based and airborne measurements of pressure, CPC, and POPS show expected relationships (at least what can be seen from the figures).

In the summary, it would be useful to state the size of the dataset, the format (netCDF), quality-control level, and other important issues (e.g., measurement biases) that users need to take into account when using this data set.

Specific comments: It would be helpful to diameter throughout the text when referring

to the size of the aerosol particles.

Line 67: when introducing the other trace gas measurements CO, NO2, SO2, O2 – the authors need to immediately state these measurements are not included as their concentrations were below the detection limit. I suggest moving lines 73 to 75 to line 67.

Line 72, Was the Gelman filter was added to avoid contamination of aerosol particles in the optical path? Specify the type of Gelman filter.

Line 133: Specify 'These variables'

Lines 127-133 and Lines 144-149 are nearly identical – I suggest combining and stated that these parameters are the same for both PRKL1 and PRKL2 copters.

Line 155: change 'Further' to 'Furthermore'

Line 160: Was the GMP343 data corrected based on the intercomparison to the PI-CARRO? A few lines later, the authors state that GMP343 suffered from inaccurate pressure compensation. Consequently, the authors recommend the use of the Licor 840A data for CO2 measurements. If this is the case, then why publish the GMP343 dataset?

Line 185: The POPS recorded 16000 cm-3 during passages of farm vehicles, which is well above the maximum concentration limit stated in Table 1. Can the POPS and N2 number concentrations be corrected for high particle concentrations?

Line 197: The authors write 'preliminary quality control' – are future updates / data products expected? The datasets published to ESSD should be better than 'preliminary'.

Line 212: change 'anc' to 'and'

Line 218: change 'written' to 'wrote'

Figure 2F shows systematic biases in the CO2 measurements (as mentioned in a comment above). Why not correct the copter CO2 measurements to the reference Picarro instrument for the final data set?

Figure 3: The times corresponding to the ground-based averages are centered at 0:00, which does not correspond to an average of the times reported in Table 5.

To facilitate comparison of the aerosol measurements in Figures 2 and 3, I suggest combining measurements of the CPCs, OPC-N2 and POPS on a single semi-log plot.

Table 1 needs to specify the instruments used for the ground-based measurements. 'Diameter' needs to be added to the size range.

---

## Referee Comment (RC3) · Anonymous Referee #3 · 16 Dec 2020

General Comments: The manuscript by Brus et al. presents a summary of the data registered during the the Lower Atmospheric Process Studies at Elevation - a Remotely-piloted Aircraft Team Experiment (LAPSE-RATE) by the Finnish Meteorological Institute (FMI) and the Kansas State University (KSU) team. This campaign was conducted in the San Luis Valley of Colorado (USA) during July of 2018. Data collected with small Unmanned Aerial systems (sUAS) and ground-based instrumentation includes aerosol particle number concentrations and size distributions, concentrations of $CO_2$ and water vapor, and meteorological parameters. This review is exclusive of the material included in the manuscript and it is not an analysis of the datasets cited in line 11. This appears to be a straightforward manuscript to read but there are some major concerns in the

specific comments below. Once the revision appropriately addresses all comments, the final manuscript should be evaluated again.

Specific Comments: 1) Because many other teams participated in the campaign and collected similar parameters, it is recommended to make the title specific to the participating teams in this manuscript (FMI-KSU flight team as mentioned in line 34). 2) l. 4: Define "FMI" here and remove it from line 35. 3) l. 5: Define "KSU" here. 4) l. 38: Define SLV. 5) l. 63 and l. 82: The manuscript should mention the thickness of the polylactide (PLA) foam cover to protect the sensors from solar radiation and display an image. This material is not really protective from the photons of the sun. The statement that "... the surface sensor module was covered from all sides with PLA foam to protect sensors from solar radiation ..." is also questionable as it would be expected that at least a percentage of photons should have made it through the PLA layer. 6) l. 64-69 and l. 153-154: From the cited paper by Barbieri et al. can be concluded that if the BME280 sensor was not forcefully aspirated, the measurements are not reliable (and there is a lag). This appears highly problematic for the work in the manuscript to provide valid data. The pressure, temperature, relative humidity sensors mounted in the second rotorcraft suffers the same problem. In conclusion, the applied compensation to the Vaisala GMP343 sensor for pressure, temperature, relative humidity (obtained from the BME280 sensor) in the postprocessing step for reporting data will only yield invalid data (for example data in Figures 2 and 3, which captions should clearly indicate the sensors used). The information in Table 1 is not valid for the BME280 sensor as mounted and operated in this work. 7) l. 89-90: Details of the custom electronics should be provided here for reproducibility. 8) l. 156: The laboratory calibration for both $CO_2$ sensors needs to be disclosed in this manuscript. 9) l. 20-23: There are a number of relevant publications in this sUAS research that should be included here to diversify the reference list and expand it from the work of the authors. The authors are encouraged to check for other new literature to be covered in this part of the introduction. 10) l. 186-187: Similar arguments for reporting only data in the ascent direction have been reported by others but have not been referenced here. 11) Figure 1 should

include in panels A and B a reference line to indicate length.

---

## Author Response (AR1)

We thank reviewer #1 for constructive comments, we really appreciate their time spent reading our manuscript.

*RC1) The manuscript by Brus et al. summarizes atmospheric data collected with UAS and ground-based sensors during the LAPSE-RATE campaign in Colorado in 2018. This is a rich dataset that will likely be mined for many years to come. The introduction would benefit from the inclusion of specific objectives. The summary would benefit from a discussion of what insights these data might provide, what questions might be addressed and answered, and where might future work be headed. Some ideas are presented below for improving and condensing the tables such that the data repository may be better linked to the manuscript.*

AR1) The introduction and summary were revised.

*RC2) L7, this sentence is a bit awkward. Let the data speak for themselves. No need to state in the abstract that things were reliable and scientifically sound. . .*

AR2) The sentence was restated in revised manuscript.

*RC3) L34, the acronyms should be listed after they are defined.*

AR3) Acronyms were listed in revised manuscript.

*RC4) L40-45, this paragraph would benefit from the inclusion of (a) specific objectives and*

*(b) specific hypotheses to be tested.*

AR4) Was restated in revised version of manuscript.

*RC5) L47, are should be were, is should be was. careful of tense throughout the paper. Past tense describing work that has been done.*

AR5) Was corrected in revised manuscript.

*RC6) L55-60, consider adding information on particle size bins and sampling rates for the*

*particle counters.*

AR6) Information was added.

*RC7) L73, neither should be for.*

AR7) Corrected.

*RC8) Section 2 should provide information on the placement of the sensors and how this placement was designed to minimize impacts of prop wash. This reviewer is particularly concerned about this salient point, given that it does not appear that the sensors were mounted above the rotary wing airframes (e.g., Nolan et al., 2018; https://www.mdpi.com/1424-8220/18/12/4448).*

AR8) The sensors of both modules (aerosol and gas) were mounted in between landing gears since the module size and weight does not allow mounting above the plain of propellers. However, in aerosol module each of the CPCs used 30 cm inlet made of conductive tubing, led upwards to the center of the

rotorcraft where both lines were merged to additional 10 cm piece of conductive inlet tubing, also facing upwards. OPC-N2 was used with no additional inlet, this OPC is not meant to be used with any kind of inlet due to use of fan for aerosol intake. On the rotorcraft the OPC was mounted from the bottom and middle of the carbon plate of the module, thus shielded from airmass movement and propeller eddies. For the gas module the influence of propellers is not of any concern.

This was clarified in revised version of manuscript.

*RC9) L95-101, provide some citations for this information please.*

AR9) Reference added.

*RC10) L140, I see sensor sampling rates in section 4, but I think these details would be better suited to a prior section. Or maybe even consider a separate section under a heading of 'sensors' where these could be separated from the description of the UAS platforms?*

AR10) The sensors sampling rate was moved to section 2. Section 2 was renamed to "Description of Platforms, Modules and Sensors".

*RC11) L184, . . .likely caused by farm vehicles. . .*

AR11) Not likely but certainly. All passing vehicles were recorded to a notebook with corresponding times.

*RC12) L187, suffered Figure 1, B. The blue flag waypoints are very hard to see in the image. Figure 3 legend, make sure to add information that this was from the car mount, approximately 2m AGL, at least as I understand it.*

AR12) Corrected accordingly.

*RC13) Table 1 is beautiful! This supports the idea of a separate section on just 'sensors', highlighting this table.*

AR13) We would rather like to keep the sensors description together with description of modules and platforms. Since the sensors are clustered to modules based on their purpose. However, the Section 2 was renamed to "Description of Platforms, Modules and Sensors".

*RC14) Tables 2,3, and 4 could be combined into a single UAS measurements table, with a platform designation column and a mission column. It would also be nice to have a flight/mission number listed here that would correspond to a labeled dataset. This will make it much easier for other folks to actually use these data.*

AR14) Tables were combined and column "mission" was added. The labeling of data sets corresponds the columns in table 2 (now merged 2.3 and 4), platform-date-UTCtime, please see detail in de Boer et al. (2020), in our opinion there is no need to introduce "a flight number".

*RC15) Tables 5 and 6 could be combined into a single surface measurements table, with just a platform column. It would also be nice to have a sampling period/mission number listed here that would correspond to a labeled dataset. This will make it much easier for other folks to actually use these data.*

AR15) Table 5 and 6 were merged as suggested by reviewer.

References:

de Boer, G., Houston, A., Jacob, J., Chilson, P. B., Smith, S. W., Argrow, B., Lawrence, D., Elston, J., Brus, D., Kemppinen, O., Klein, P., Lundquist, J. K., Waugh, S., Bailey, S. C. C., Frazier, A., Sama, M. P., Crick, C., Schmale III, D., Pinto, J., Pillar-Little, E. A., Natalie, V., and Jensen, A.: Data generated during the 2018 LAPSE-RATE campaign: an introduction and overview, Earth Syst. Sci. Data, 12, 3357–3366, https://doi.org/10.5194/essd-12-3357-2020, 2020.

Anonymous Referee #2

*Review of Atmospheric aerosol, gases and meteorological parameters measured during the LAPSE-RATE campaign, D. Brus et al.*

We thank reviewer #2 for constructive comments, we really appreciate their time spent reading our manuscript.

This manuscript presents an overview of the data acquired from copter UAVs that conducted vertical profiles of aerosols, gases and meteorological parameters during the LAPSE-RATE campaign in the summer of 2018. Members from the Finnish Meteorological Institute (FMI) and Kansas State University (KSU) conducted measurements over a period of five days in the San Luis Valley, CO. The FMI copter measurements include vertical profiles up to nearly 900 m AGL of aerosol number concentrations and size distributions, $CO_2$, and meteorological parameters (pressure, temperature, relative humidity), while KSU copter measurements conducted vertical profiles (up to 150 m AGL) of aerosol size and number distributions. In addition, FMI and KSU conducted ground-based measurements of aerosol and meteorological parameters to compare with the airborne copter measurements. The data from these flights is readily available from the websites provided in the manuscript. It is understood that ESSD is dedicated to the publication of datasets; however, key features of the dataset as well as its limits are needed to highlight the utility of the data in future publications. Several issues are described below that the authors should address before publication.

General comments:

*RC1) The authors allude to a number of regional / local sources and meteorological patterns that impact diurnal cycles, changes in aerosol properties, generation of new particle formation events. However, few specific examples were highlighted in the text (farm vehicles), and summary (new particle formation). A few sentences on the defining characteristics of this dataset (for example, temporally with respect to meteorological patterns and vertically with respect to atmospheric structure) would be useful.*

AR1) In general, there was/is very little known about the background aerosol concentrations within the SLV from the literature. The aerosol and generally any air quality measurements are very sparse in Colorado and are mostly concentrated around larger cities. The analysis of the data collected by FMI and KSU, however limited, with narrative is provided in our ACP publication Brus et al. 2021. We will provide several sentences to reflect reviewer's comment.

*RC3) Airborne aerosol measurements are a challenge – especially for measurements of particles larger than several micrometers in diameter in a non-isokinetic flow. A description of the KSU inlet has been provided; however, the orientation with respect to the wind and propeller wash is not clear. The stated largest diameter for the N2 and POPS are 17 and 33 um, respectively. The authors need to provide an assessment of sampling biases related to super-micron aerosol particles. I also suggest that authors compare ground-based and airborne measurements for the OPC-N2 and POPS at a range of sizes between 0.3 and 30 um diameter.*

AR3) The inlet orientation of POPS was clarified in revised manuscript. There was a terrible mistake in decimal point on line 186 concerning the mid bin diameters of POPS, however in many other places it is clearly stated that the lowest and largest diameter measurable by POPS are 0.13 and 3.65 microns. OPC-

N2 and POPS overlap in the range between 0.46 to 3.5 microns. Discussion on sampling losses and comparison of airborne and ground measurements was also added.

The following was added to line 63:" Each of the CPCs used a 30 cm inlet made of conductive tubing, led from the sides upwards to the center of the rotorcraft, where both lines were merged to an additional 10 cm piece of conductive inlet tubing, also facing upwards. Penetration for such an inlet was estimated to be between 90% and 99% or particles between 7 and 100 nm and 99 % for particles between 100 nm and 1um."

To line 84:"Both OPCs N2, in the particle and ground module, were used with no additional inlet, as those OPCs were not meant to be used with any kind of inlet due to the use of a fan for aerosol intake."

To line 96:" … a horizontally oriented naked inlet approximately 9 cm (3.5 in.) long with an inner diameter of 1.7 mm (0.069 in.)."

To line 100: "… it was located approximately 1.8 m AGL and used a vertically oriented tube inlet of approximately 45cm (18 in.) long with an inner diameter of 3.175 mm (0.125 in.). The penetration through the inlet was estimated to be ~92% for 3 um particles and better for smaller ones."

*RC4) Along the same line, when comparing the OPC-N2 between the copter and the ground based measurements (Figures 2D,E and Figures 3C,D), it appears that the N2 concentrations on the copter are systematically at least a factor of two larger than the ground based measurements. Was there any additional flow control or flow measurements for the OPC-N2? The off-the-shelf version does not provide precise measurement of the air volume for determining the number concentration.*

AR5) No, we did not make any additional flow measurements or additional flow control of OPC-N2, it was used as it was purchased. The nominal flow through OPC-N2 is ~220 ccm/min and is calculated with a time-of-flight method. Transit times of particles are recorded to eventually correct the flow speed. The comparison between surface and airborne measurements please see AR6 below.

*RC5) The authors state that hysteresis between ascent and descent profiles was significant. This difference is expected given the high ascent rates (up to 8 m. s-1) and considerably slower descent rates (as low as 2 m.s-1). The authors then suggest using the ascents for the best representation of the vertical profiles with no justification. Given that the ascent rates were faster, the impact of hysteresis on the vertical profile should also be greater. Are there other factors that need to be considered? What is the bias related to the hysteresis? I also suggest showing a figure to illustrate the impact of hysteresis on the vertical profiles.*

AR5) The authors state the hysteresis was noticeable. The ascent data are recommended for use since they copy the temperature profile better when compared to sounding data from Leech airport located about 15 km from the FMI and the KSU teams sampling location. As an example, please see attached FIG. 1 and FIG. 2 for comparison profiles of temperature and RH at matching times between multicopter and sounding. The BME280 sensor descending profiles are flatter compared to ascend profiles, ascend data copy the sounding slope better in our opinion. It seems the sensor cools faster than it warms up, since the observed hysteresis. In our experience, when the sensor was flown against reference 300 m tower and the ascend and descent rates are close to 1 m/s the hysteresis disappears. But during LAPSE-RATE such slow ascend and descent rates were in contrary to our preference to sample aerosol

properties in whole permitted vertical column 3000 feet, i.e. climb very fast in a limited time given by multicopter battery capacity.

The profiles are plotted separately in our ACP manuscript Brus et al. (2021) with detailed discussion, here we would like to provide solely data sets.

[Figure]

Figure 1. Comparison of temperature profiles between BME280 on multicopter and sounding made at Leech airport.

[Figure]

Figure 2. Comparison of RH profiles between BME280 on multicopter and sounding made at Leech airport.

*RC6) The figures show time series of averaged values and variability for each flight. However, there are no examples of vertical profiles and no specific comparisons / validations between airborne and ground-based data. Yet, there are some differences between the airborne and ground-based averages that cannot be reconciled in the figures that have been shown. For example, ground-based temperature and relative humidity show no consistent relationship with the airborne observations. I would have expected to see the ground-based temperature similar to the warmest airborne temperature in a wellmixed boundary layer. As mentioned above, the ground-based number concentrations reported by the OPC N2 are consistently less than the airborne values by more than a factor of two. Otherwise, ground-based and airborne measurements of pressure, CPC, and POPS show expected relationships (at least what can be seen from the figures).*

A very similar comment was raised during the review of our manuscript in ACP Brus et al. (2021) the following analysis was provided, and it can be accessed via Supplementary materials of the above mentioned manuscript.

We performed a short comparison (about 5 minutes) before each flight among the particle counters to check their performance, mostly visually from the laptop screen, if the number concentrations roughly correspond to each other. The rotorcraft with particle module was not in the same location as surface module, neither the particle counters were using the same inlet. The rotorcraft was standing on the camping table approximately one third of the height of surface module which was placed on the car roof. Since the provided comparison is rather semi quantitative.

The comparison of rotorcraft particle module to surface module for CPC could be seen in attached FIG. 3, we must point out that each CPC was calibrated to different D50 cut-off, the most pronounced disagreement could be seen on Jul 16[th] when a weak NPF took place also at the surface (the red circles).

[Figure]

Figure. 3 Inter-comparison of CPCs mounted on rotorcraft particle module (CPC1 and CPC2) and surface module (CPC Ground).

The comparison of OPCs in particle and surface module is shown in attached FIG. 4. In some cases, the OPC on particle module shows higher concentrations than the OPC on surface module. This might due to rotorcraft proximity to dusty surface during comparison. Similarly, when we compare normalized concentration per bin, the OPC on particle module slightly overcounts in all bins, see FIG. 5.

[Figure]

Figure 4. Daily comparison of total number concentration of OPCs mounted on rotorcraft particle module (OPC-N2_RC) and surface module (OPC-N2_Surface).

[Figure]

Figure 5. Daily comparison of normalized concentration per bin of the OPCs mounted on rotorcraft particle module (OPC-N2_RC) and surface module (OPC-N2_Surface).

There was no intentional comparison made for the pair of POPS counters, however we made a comparison of total particle number concentration using the air unit data just before the flight, when the KSU rotorcraft was ready for take-off, e.g. height was zero or close to zero, see Fig 6. The particle concertation data are slightly biased toward higher counts of air unit, on average about 10%.

[Figure]

Figure 6. Comparison of POPS air and surface unit particle number concentration.

We will add following to manuscript, line 209: "Also, a short inter-comparison (~5 minutes) was performed before each flight among the surface and airborne particle counters to check their performance. Based on data postprocessing, the CPCs of particle and surface modules compared well within the manuscript stated uncertainty of 10%, except for July 16[th] when NPF at the surface level took place. This is due to different calibrated cut-off diameter of each CPC. The OPCs compared within factor of 2, however it has to be considered that very low particle concentrations were measured, about 2 cm[-3] in the OPCs size range. There were no inter-comparison measurements made for POPS instruments. For more details please see Supplementary materials of Brus et al. (2021), where the detailed analysis was provided."

*RC7) In the summary, it would be useful to state the size of the dataset, the format (netCDF), quality-control level, and other important issues (e.g., measurement biases) that users need to take into account when using this data set.*

AR7) The following was added to Summary section: The dataset was divided into two parts: FMI dataset containing 60 files (21 MB) and KSU dataset containing 31 files (11.3 MB), all files are available in netCDF

format. The QC and measurement biases are discussed in details in appropriate sections of the manuscript.

Specific comments:

*RC8) It would be helpful to diameter throughout the text when referring to the size of the aerosol particles.*

AR8) Corrected accordingly.

*RC9) Line 67: when introducing the other trace gas measurements CO, NO2, SO2, O2 the authors need to immediately state these measurements are not included as their concentrations were below the detection limit. I suggest moving lines 73 to 75 to line 67.*

AR9) Corrected according reviewer's suggestion.

*RC10) Line 72, Was the Gelman filter was added to avoid contamination of aerosol particles in the optical path? Specify the type of Gelman filter.*

AR10) Yes, the filter is included to protect the optical path from contamination. Producer (Li-cor) recommended and provided Gelman 1 Micron Filter Assembly was used. The manuscript was updated accordingly.

*RC11) Line 133: Specify 'These variables' Lines 127-133 and Lines 144-149 are nearly identical – I suggest combining and stated that these parameters are the same for both PRKL1 and PRKL2 copters.*

AR11) Changed according to reviewer suggestion.

*RC12) Line 155: change 'Further' to 'Furthermore'*

AR12) Changed accordingly.

*RC13) Line 160: Was the GMP343 data corrected based on the intercomparison to the PICARRO? A few lines later, the authors state that GMP343 suffered from inaccurate pressure compensation. Consequently, the authors recommend the use of the Licor 840A data for CO2 measurements. If this is the case, then why publish the GMP343 dataset?*

AR13) No, the data were not corrected for the bias after measurements against Picarro. The correction to our dataset cannot be done since the comparison to Picarro was done at sea level and our dataset from GMP343 is obtained in vertical column i.e. changing pressure. Our point is that the Vaisala proprietary compensation algorithm does not work well for such application, the device (GMP343) is simply not meant for UAV vertical profiling. On the other hand, the GMP343 when compared to Li-840A has very nice shape factor that makes it suitable for use in for example a small fixed wing UAV and sampling plumes, however lower precision of the GMP343 has to be kept in mind.

The GMP343 data set is published for the case there would be someone interested in developing better compensation algorithm for this probe, all data necessary to do it are included in our dataset.

*RC14) Line 185: The POPS recorded 16000 cm-3 during passages of farm vehicles, which is well above the maximum concentration limit stated in Table 1. Can the POPS and N2 number concentrations be corrected for high particle concentrations?*

AR14) Such data were cleaned off from the data sets, our interest was only in characterizing background aerosol concentrations in SLV and not an exhaust emission of passing cars. The coincidence counts are in place in case of POPS, in Mei et al. (2020) they claim that the coincidence error is less than 25% when the concentration is less than 4000/ccm when compared to Ultra-High Sensitivity Aerosol Spectrometer (UHSAS). The car exhaust particles are smaller than the range of OPC-N2, thus OPC-N2 was not suffering from the coincidence counts. When measuring aerosols in plumes or exhausts the dilution of sampling flow is preferable over coincidence count corrections, as mentioned above this was not our intention during this campaign.

*RC15) Line 197: The authors write 'preliminary quality control' – are future updates / data products expected? The datasets published to ESSD should be better than 'preliminary'.*

AR15) The word "preliminary" was omitted from the sentence. It was not meant as it was understood, our apology for confusion, all data sets were quality checked. We published our limited analyses of the data sets already in Brus et al. (2021) and the authors currently have no other planes on creating further products out of those data sets.

*RC16) Line 212: change 'anc' to 'and'*

AR16) Corrected accordingly.

*RC17) Line 218: change 'written' to 'wrote'*

AR18) Corrected accordingly.

*RC18) Figure 2F shows systematic biases in the CO2 measurements (as mentioned in a comment above). Why not correct the copter CO2 measurements to the reference Picarro instrument for the final data set?*

AR18) Answered in a comment above.

*RC19) Figure 3: The times corresponding to the ground-based averages are centered at 0:00, which does not correspond to an average of the times reported in Table 5.*

AR19) The mistake was corrected.

*RC20) To facilitate comparison of the aerosol measurements in Figures 2 and 3, I suggest combining measurements of the CPCs, OPC-N2 and POPS on a single semi-log plot.*

AR20) Combining all three instruments into a single plot completely flattens the data and hides the daily variations, even though it is not very pronounced. The three orders of magnitude between CPC+POPS and OPC-N2 is too huge. We would like to keep separate figures as they are.

*RC21) Table 1 needs to specify the instruments used for the ground-based measurements. 'Diameter' needs to be added to the size range.*

AR21) Changed accordingly.

References:

Brus, D., Gustafsson, J., Vakkari, V., Kemppinen, O., de Boer, G., and Hirsikko, A.: Measurement report: Properties of aerosol and gases in the vertical profile during the LAPSE-RATE campaign, Atmos. Chem. Phys., 21, 517–533, https://doi.org/10.5194/acp-21-517-2021, 2021.

Mei F, McMeeking G, Pekour M, Gao R-S, Kulkarni G, China S, Telg H, Dexheimer D, Tomlinson J, Schmid B. Performance Assessment of Portable Optical Particle Spectrometer (POPS). Sensors. 2020; 20(21):6294. https://doi.org/10.3390/s20216294

Anonymous Referee #3

We thank reviewer #3 for constructive comments, we really appreciate their time spent reading our manuscript.

General Comments:

The manuscript by Brus et al. presents a summary of the data registered during the Lower Atmospheric Process Studies at Elevation - a Remotely piloted Aircraft Team Experiment (LAPSE-RATE) by the Finnish Meteorological Institute (FMI) and the Kansas State University (KSU) team. This campaign was conducted in the San Luis Valley of Colorado (USA) during July of 2018. Data collected with small Unmanned Aerial systems (sUAS) and ground-based instrumentation includes aerosol particle number concentrations and size distributions, concentrations of CO2 and water vapor, and meteorological parameters. This review is exclusive of the material included in the manuscript and it is not an analysis of the datasets cited in line 11. This appears to be a straightforward manuscript to read but there are some major concerns in the specific comments below. Once the revision appropriately addresses all comments, the final manuscript should be evaluated again.

Specific Comments:

*RC1) 1) Because many other teams participated in the campaign and collected similar parameters, it is recommended to make the title specific to the participating teams in this manuscript (FMI-KSU flight team as mentioned in line 34).*

AR1) The title was changed to:" Atmospheric aerosol, gases and meteorological parameters measured during the LAPSE-RATE campaign by Finnish Meteorological Institute and Kansas State University"

*RC2) l. 4: Define "FMI" here and remove it from line 35.*

AR2) Changed accordingly

*RC3) l. 5: Define "KSU" here.*

AR3) Changed accordingly

RC4) l.38: Define SLV.

AR4) Defined in Abstract, line 2.

*RC5) l. 63 and l. 82: The manuscript should mention the thickness of the polylactide (PLA) foam cover to protect the sensors from solar radiation and display an image. This material is not really protective from the photons of the sun. The statement that ". . . the surface sensor module was covered from all sides with PLA foam to protect sensors from solar radiation . . ." is also questionable as it would be expected that at least a percentage of photons should have made it through the PLA layer.*

AR5) The sentences were changed as follows: "… module was covered from all sides except the bottom with a polylactide (PLA) foam cover (2.5 cm thick) to shade the sensors from direct sun and keep the particle module thermally stable."

*RC6) l. 64-69 and l. 153-154: From the cited paper by Barbieri et al. can be concluded that if the BME280 sensor was not forcefully aspirated, the measurements are not reliable (and there is a lag). This appears*

*highly problematic for the work in the manuscript to provide valid data. The pressure, temperature, relative humidity sensors mounted in the second rotorcraft suffers the same problem. In conclusion, the applied compensation to the Vaisala GMP343 sensor for pressure, temperature, relative humidity (obtained from the BME280 sensor) in the postprocessing step for reporting data will only yield invalid data (for example data in Figures 2 and 3, which captions should clearly indicate the sensors used). The information in Table 1 is not valid for the BME280 sensor as mounted and operated in this work.*

AR6) In Barbieri et al. 2019 it was discussed that placement of the sensor (shielded/not-shielded, aspirated/not-aspirated) plays dominant role rather than its own accuracy and uncertainty, that was also valid in our case, sensors shielded but not forcefully aspirated. When BME280 sensors were calibrated in environmental chamber – well controlled conditions, against national standard at FMI, their response was very satisfactory, please see FIG 1.

[Figure]

[Figure]

Figure 1. T and RH calibration for 9 pcs of BME280 sensors against FMI national standard.

Yes, we compensated the GMP343 with temperature, RH and pressure obtained from the BME280 sensor that showed a bias of about +2 C in temperature, −12 % in RH and +2 hPa in pressure during the LAPSE-RATE inter-comparison measurements against MURC (Barbieri et al., 2019), we also accounted for that bias in compensation. Even when accounting for the maximum error the resulting change (increase) in $CO_2$ concentration was only 2 ppm.

Table 1 caption was updated to "Table 1. An overview of sensors and their operational characteristics provided by manufacturer…."

*RC7) l. 89-90: Details of the custom electronics should be provided here for reproducibility.*

AR7) The word "custom" was omitted; it was used inaccurately in this context.

*RC8) l. 156: The laboratory calibration for both CO2 sensors needs to be disclosed in this manuscript.*

AR8) The following sentence was added to line 174: "The following laboratory-derived calibration constants were applied to the datasets collected by both sensors:
Licor_new=0.95785×Licor_measured+8.66055 with R^2=0.9999 and

GMP_new=0.99878×GMP_compensated+8.77014 with R^2=0.99999. Please note GMP_compensated used in the equation above and that the Vaisala compensation algorithm is confidential."

*RC9) l. 20-23: There are a number of relevant publications in this sUAS research that should be included here to diversify the reference list and expand it from the work of the authors. The authors are encouraged to check for other new literature to be covered in this part of the introduction.*

AR 9) The references were updated.

*RC10) l. 186-187: Similar arguments for reporting only data in the ascent direction have been reported by others but have not been referenced here.*

AR10) We reference here to reviewer #2 comment 5) and our answer in there.

*RC11) Figure 1 should include in panels A and B a reference line to indicate length.*

AR11) Figure 1 was updated accordingly.

*Topical Editor and Discussion Comments for*

*"Atmospheric aerosol, gases and meteorological parameters measured during the LAPSE-RATE campaign"*

*The paper presents data sets of atmospheric and surface aerosols and gases measured by researchers from the Finnish Meteorological Institute (FMI) and Kansas State University (KSU) during the Colorado San Luis Valley LAPSE-RATE flight campaign. Vertical profiles were flown with small unmanned aerial vehicles (sUAS) carrying payloads comprised of multiple sensing instruments. Details of the instruments, measurements, and operations are included in the paper.*

Thank you Suzanne for careful reading!

*TC1) pg1, lines 4-5: define (FMI) and (KSU) here for use in the next paragraph*

AR1) Changed accordingly.

*TC2) pg 1, line 17: possible extra space between "mostly  on"*

AR2) There was no extra space, just Latex formatting.

*TC3)  pg 2, line23: used "sUAS" in abstract and "UASs" here; make consistent*

AR3) in this context it is not small UAS but plural for UAS, e.g. Ramanathan et al (2007) used pretty big fixed wings.

*TC4) pg 2, line29: should the de Boer reference be 2020a (not 2018)?*

AR4) True, corrected accordingly.

*TC5) pg 2, line 35: if defined in abstract, (FMI) and (KSU) definition can be removed here*

AR5) Corrected accordingly.

*TC6) pg 2, line 41: possibly specify time of year, "… substantial summer diurnal cycles …"*

AR6) Corrected accordingly.

*TC 7) pg 3, line73: sentence issue, possibly replace "neither" with "nor"?*

AR7) Replaced.

*TC8) pg4, line 93: This sentence referring to Table 1 would be more useful for the reader if it preceded the sensor description text, possibly added at the end of pg3, line 54.*

AR8) The sentence was moved accordingly.

*TC9) pg 4, line114: As redundant flights were conducted to evaluate instrument consistency, an additional comment about the consistency would be appreciated, wither here or note that it is included later with other discussion of results.*

AR 9) Following sentences were added: "It was found that POPS was counting consistently during vertical sampling. But during the horizontal transects, when the POPS inlet was facing the direction of flight, we found fluctuations in POPS measured sample flow rate. Only vertical profile data are included in the dataset."

*TC10) pg 6, lines 155 and 187-188: You advise the reader of potential issues with the proprietary Vaisala compensation algorithm. Can you add an indication of how this advice was realized, e.g., via your laboratory calibration or through comparison to redundant flight sensing or other?*

AR10) The GMP sensor performs quite well when compared to LI-Cor 840A and higher precision Picarro instrument at sea level in Helsinki (p.7, l193). That does not hold when compared to Li-Cor in SLV (2300 m MSL) and gets even worse in vertical profile. Since we are pointing to pressure compensation algorithm.

*TC11) pg 6, line 182-183: Possibly include a subheading to signal a shift to discussion of the data.*

AR11) Subsection inserted.

*TC12) pg7, line 190: Suggest adding a note to draw the reader's attention to something to notice in the data. Also, are the error bars or other aspects of the results comparable to those of prior atmospheric studies with sUAS?*

AR12) For interested reader we provide analysis of our dataset in Brus et al 2021.

*TC13) pg 7, line 190: Several other papers in the collection also present a focused one-day data set showing temporal or altitude data variations seen with the sUAS instrumentation. I realize that you also are publishing results in Brus 2020c, so that could be noted here as well.*

AR13) the same as AR12.

*TC 14) pg 7, line 197: You mention minimal quality control a few places and that data were as recorded, but you also describe lab calibration tests of the instruments before and after the campaign. Do I understand correctly that you were satisfied that the instruments were working well by the lab validation, but did not apply any calibration adjustment to the data set? The multiple mentions about the data processing (or lack of it) seemed inconsistent and led to some confusion. If the processing is different depending on the instrument, a column could be added to Table 1 to distinguish.*

AR 14) Similar sentences like this "All of these data are published as they were measured without any corrections or additional quality control." appear through the manuscript, in this case it only refers to attitude data of flight controller, GPS, and Meteorological parameters. In cases were the correction and calibration were applied (gas sensors) it is particularly mentioned. I am sorry for confusion I have omitted those sentences or tried to clarify.

*TC 15)*
*pgs 9-12: The following notes pertain to the references:*
- *It was a quick check, but 19 of the references were not referred to in the text.*
- *Jonassen, on pg 11, is not in alpha order.*
- *Need to update status of Brus 2020c, Bell 2020, de Boer 2020a, 2020b, and 2020c*
- *Inconsistent inclusion of reference-ending period.*

AR15) Thanks for careful check, I forgot to comment out those extra references. References were updated and cleaned up.

*TC16) pg13, subplot D: The CPC1 points on 18 Jul set the scale for this plot, reducing the visibility of the other data. Are these results something to note in the text or in the caption so that the rest of the data can be seen more clearly?*

AR16) Note to figure caption was added, "…D) total aerosol number concentration by CPCs and POPS, note the elevated concentration measured by CPC1 characterizing the NPF event, E)…"

*TC17) pg 16-17, Tables 2 and 3: Need units (m MSL and m ASL) added to two righthand columns headings or to Table caption. Horizontal date dividers (like Table 4) would be useful here as well.*

AR17) Corrected accordingly.

*TC18) pg 19, Tables 5 and 6: The reader would be helped if the caption included the ground instrumentation altitude (2291 m MSL).*

AR18) altitude added to caption.